# Control of SARS-CoV-2 infection after Spike DNA or Spike DNA+Protein co-immunization in rhesus macaques

Margherita Rosati[1], Mahesh Agarwal[2], Xintao Hu[2], Santhi Devasundaram[2], Dimitris Stellas[1], Bhabadeb Chowdhury[1], Jenifer Bear[2], Robert Burns[2], Duncan Donohue[3], Laurent Pessaint[4], Hanne Andersen[4], Mark G. Lewis[4], Evangelos Terpos[5], Meletios Athanasios Dimopoulos[5], Alexander Wlodawer[6], James I. Mullins[7,8,9], David J. Venzon[10], George N. Pavlakis[1], Barbara K. Felber[2]*

1 Human Retrovirus Section, Vaccine Branch, Center for Cancer Research, National Cancer Institute at Frederick, Frederick, Maryland, United States of America, 2 Human Retrovirus Pathogenesis Section, Vaccine Branch, Center for Cancer Research, National Cancer Institute at Frederick, Frederick, Maryland, United States of America, 3 MS Applied Information and Management Sciences, Frederick National Laboratory for Cancer Research, Frederick, Maryland, United States of America, 4 BIOQUAL, Inc.; Rockville, Maryland, United States of America, 5 Department of Clinical Therapeutics, National and Kapodistrian University of Athens, School of Medicine, Athens, Greece, 6 Center for Structural Biology, National Cancer Institute, Frederick, Maryland, United States of America, 7 Department of Microbiology, University of Washington, Seattle, Washington, United States of America, 8 Department of Medicine, University of Washington, Seattle, Washington, United States of America, 9 Department of Global Health, University of Washington, Seattle, Washington, United States of America, 10 Biostatistics and Data Management Section, Center for Cancer Research, National Cancer Institute, National Institutes of Health, Bethesda, Maryland, United States of America

* Barbara.felber@nih.gov

**Data Availability Statement:** All relevant data are within the manuscript and its Supporting information files.

## Abstract

The speed of development, versatility and efficacy of mRNA-based vaccines have been amply demonstrated in the case of SARS-CoV-2. DNA vaccines represent an important alternative since they induce both humoral and cellular immune responses in animal models and in human trials. We tested the immunogenicity and protective efficacy of DNA-based vaccine regimens expressing different prefusion-stabilized Wuhan-Hu-1 SARS-CoV-2 Spike antigens upon intramuscular injection followed by electroporation in rhesus macaques. Different Spike DNA vaccine regimens induced antibodies that potently neutralized SARS-CoV-2 *in vitro* and elicited robust T cell responses. The antibodies recognized and potently neutralized a panel of different Spike variants including Alpha, Delta, Epsilon, Eta and A.23.1, but to a lesser extent Beta and Gamma. The DNA-only vaccine regimens were compared to a regimen that included co-immunization of Spike DNA and protein in the same anatomical site, the latter of which showed significant higher antibody responses. All vaccine regimens led to control of SARS-CoV-2 intranasal/intratracheal challenge and absence of virus dissemination to the lower respiratory tract. Vaccine-induced binding and neutralizing antibody titers and antibody-dependent cellular phagocytosis inversely correlated with transient virus levels in the nasal mucosa. Importantly, the Spike DNA+Protein co-immunization regimen induced the highest binding and neutralizing

**Funding:** This work was supported by funding from the Intramural Research Program, National Institutes of Health, National Cancer Institute, Center for Cancer Research to A.W., G.N.P. and B. K.F., by Inovio Pharmaceuticals Inc. Plymouth Meeting, PA, under NCI CRADA#02289 to G.N.P and B.K.F., in part by the University of Washington Population Health Initiative to J.I.M., and the University of Washington Centers for AIDS Research Retroviruses and Molecular Data Sciences Core (P30 AI027757; J.I.M). The funders had no role in study design, data collection and analysis, decision to publish, or preparation of the manuscript.

**Competing interests:** The authors have declared that no competing interests exist.

antibodies and showed the strongest control against SARS-CoV-2 challenge in rhesus macaques.

## Author summary

Anti-Spike neutralizing antibodies provide strong protection against SARS-CoV-2 infection in animal models, and correlate with protection in humans, supporting the notion that induction of strong humoral immunity is key to protection. We show induction of robust antibody and T cell responses by different Spike DNA-based vaccine regimens able to effectively mediate protection and to control SARS-CoV-2 infection in the rhesus macaque model. This study provides the opportunity to compare vaccines able to induce different humoral and cellular immune responses in an effort to develop durable immunity against the SARS-CoV-2. A vaccine regimen comprising simultaneous co-immunization of DNA and Protein at the same anatomical site showed best neutralizing abilities and was more effective than DNA alone in inducing protective immune responses and controlling SARS-CoV-2 infection. Thus, an expansion of the DNA vaccine regimen to include co-immunization with Spike protein may be of advantage also for SARS-CoV-2.

## Introduction

SARS-CoV-2 has infected to date more than 167 million individuals worldwide and is responsible for more than 3.5 million deaths to date (https://www.coronavirustraining.org/live-map). Several SARS-CoV-2 vaccines, approved under Emergency Use Authorization (EUA) and Conditional Marketing Authorization (for review see [1,2] and references therein), or that are being considered [3–13], have shown strong protective efficacy against the development of disease. Anti-Spike neutralizing antibodies (NAb) provide strong protection against infection in culture and in animal models, and correlate with protection in humans, supporting the notion that induction of strong humoral immunity is key to protection [1,2].

While different vaccine platforms are being used [reviewed in [1]], it is noteworthy that nucleic acid-based vaccines (mRNA and DNA) have been the faster to develop compared to approaches using recombinant viral vectors, protein, peptides or inactivated virus. The mRNA and DNA vaccine platforms are promising due to their simplicity, scalability, and in the case of viral vectors, the possibility of repeated applications due to the lack of immunity against the vector (for reviews see [14–20]). In contrast to mRNA vaccines that induce robust antibody responses but low T cell responses, several clinical trials have shown that intramuscular DNA vaccines induce both humoral and strong cellular immune responses. The latter may be important for the induction of durable antibody responses.

DNA vaccines encoding the SARS-CoV-2 Spike protein were first to demonstrate protection in the macaque model [21]. Protective responses induced by DNA were reported also by others [22] and a Spike DNA vaccine is advancing to clinical trials [12]. SARS-CoV-2 mRNA as a nucleic acid vaccine platform also demonstrated protection from infection/disease development in animal models [23,24] and led to robust humoral immune responses in humans [3,25,26]. The mRNA-based vaccines were first to receive EUA and Conditional Marketing Authorization in USA and European Union, respectively. Alphavirus- derived replicon RNA (repRNA), an alternative nucleic acid vaccine platform, also induced strong anti-Spike humoral immune responses in the macaque model [27] and is advancing to the clinic.

We have long-standing experience in developing DNA vaccines. We have developed DNA+Protein co-immunization regimens that maximize induction of humoral and cellular immune responses. These vaccine regimens have resulted in rapid development of immune response and potent protection against infection with the Simian Immunodeficiency Virus (SIV) and hybrid Simian-Human Immunodeficiency Virus (SHIV) model in rhesus macaques [28–33]. Here, we report the testing of DNA and DNA+Protein co-immunization vaccine regimens against SARS-CoV-2. We demonstrate induction of potent protective immune responses and control of virus challenge in the Indian-origin rhesus macaque model.

## Results

### Spike DNA vaccine and immunogenicity in mice

A series of plasmid DNAs (Fig 1A) was generated to express different forms of the prefusion stabilized SARS-CoV-2 Spike, all of which included the introduced 2P stabilizing mutation [34–36]. All constructs, except the full-length S-1273, also have a furin cleavage site mutation ($_{679}$NSPRRA$_{684}$ replaced by residues I-L; denoted ΔF), reported to maintain ACE2 receptor recognition [37]. DNAs expressing Spike ΔF were designed to maximize production of prefusion Spike. Spike S1/S2_ΔF_FO comprises the extracellular domains of the Spike S1 and S2 proteins with deletions of the C-terminal region previously associated with antibody dependent enhancement of SARS [38], the membrane-spanning domain (MSD) and C-terminal cytoplasmic region, which is replaced with the T4 foldon (FO) to generate a secreted Spike in its prefusion configuration. Plasmid S-RBD_ΔF_FO expressed an extended constrained Spike receptor binding domain (S-RBD, AA 295–1171) and has two mutations, K304E and S944E, introduced to facilitate electrostatic bridges with the structurally neighboring K964 and K310 residues, respectively. This protein lacks the MSD and the C-terminal region which is substituted with the T4 foldon to constrain RBD conformation.

Upon transient transfection of HEK293 cells, all plasmids were efficiently expressed. Western blot analysis (S1A Fig) showed that the S-1273 DNA produced full-length Spike protein in the cell-associated fraction and secreted cleaved S1. All Spike proteins with the ΔF mutation, including the S-RBD, were found primarily in the cell-associated fraction. S-1273_ΔF DNA produced higher levels of the complete prefusion Spike compared to S-1273 DNA.

The immunogenicity of Spike-encoding DNAs was first tested in BALB/c mice. Mice were vaccinated at week 0 and 3, followed by antibody measurements 2 weeks later (S1B–S1D Fig). Similar levels of anti-Spike-RBD antibodies were induced by S-1273 and S-1273_ΔF DNA, reaching median AUC of 5.8 log (range 4.5–7.8) and 6.6 log (range 5.3–7.1), respectively (S1C Fig). The foldon Spike variants induced significantly lower responses with median AUC titers of 4.3 log (range 2.4–4.9) and 2.6 log (range 0.1–4.4), respectively. Neutralizing antibodies (NAb) were measured in pooled sera using the HIV-1$_{NL}$ΔEnv-NanoLuc derived pseudotype neutralization assay in HEK-ACE2 cells [39]. All vaccines induced NAb and a significant direct correlation was found between Spike-RBD binding antibody and NAb titers (S1D Fig). These data showed that vaccination by DNAs expressing S-1273 and S-1273_ΔF induced significantly higher antibody responses in mice than the foldon Spike variants.

### Spike DNA and Spike DNA+Protein vaccination of rhesus macaques

Six DNA vaccine regimens were tested in groups (G1-G6) of Indian rhesus macaques (4 animals per group, except group 2 with 8 animals), delivered by IM injection followed by electroporation following protocols that included either 3 (G1, G2, G3, G5) or 2 (G4, G6) vaccinations (Fig 1A). The vaccination schedule for G4 differed from others in that a longer 5-week rest period between prime and boost was used, instead of 3 weeks. A regimen of co-

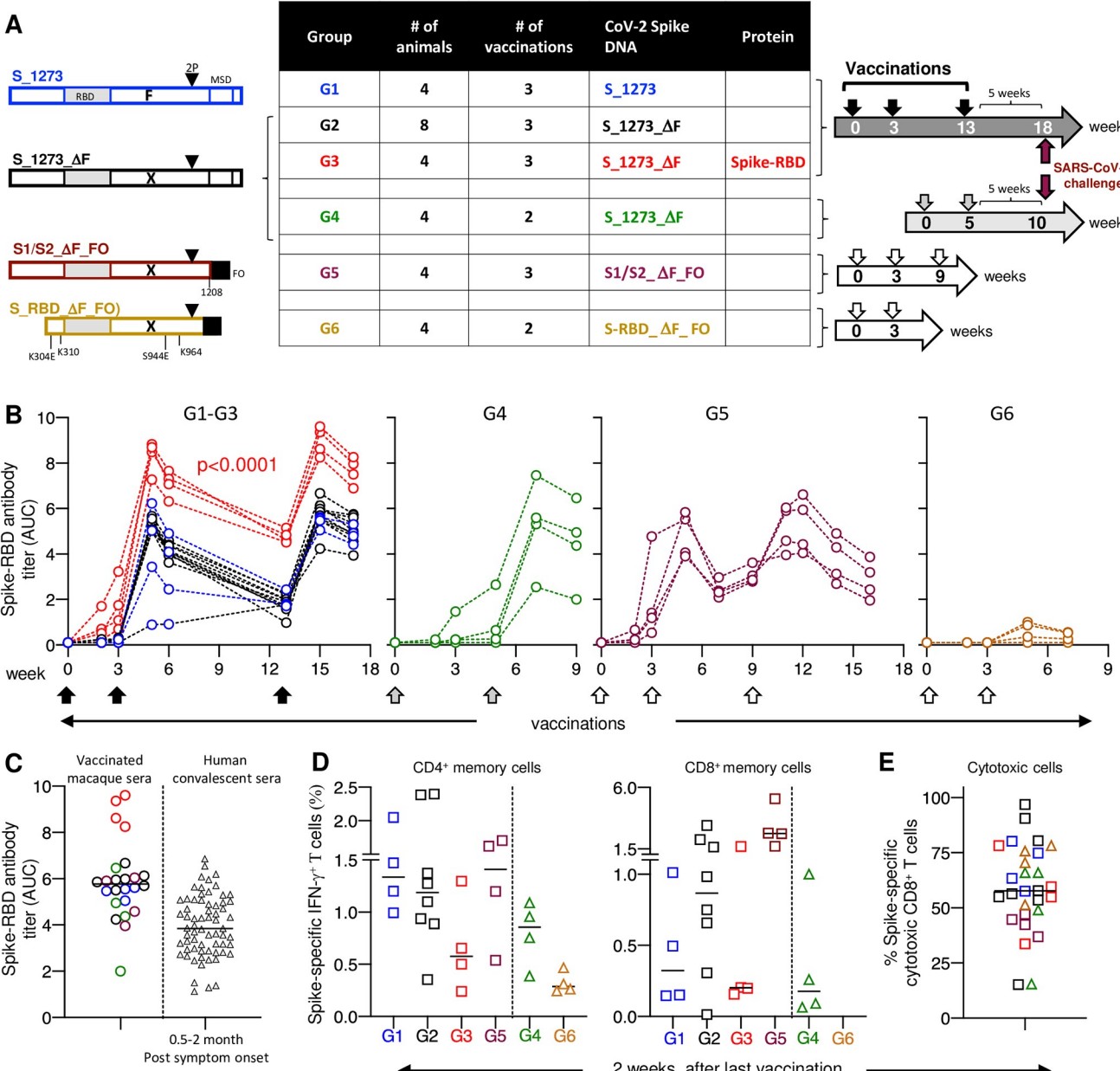

**Fig 1. Vaccination-challenge study in macaques.** (**A**) Cartoon depicts the Spike immunogens produced from the different DNA-based vaccines and the vaccination scheme of the 6 vaccine groups. The animals from G1-G4 were challenged 5 weeks after the last vaccination (indicated by maroon arrow). Blood was collected at different time points after each vaccination. (**B**) Anti-Spike-RBD antibodies were measured over time by ELISA in the vaccinated macaques and shown as AUC. P values is from ANOVA. Titers reached after the 3rd vaccination correspond to median model fit endpoint titers of 6 logs for the DNA+Protein group (G3) and 5.1 logs for the DNA-only groups (G1, G2, G5). (**C**) Anti-Spike-RBD responses from macaques after the last vaccination were compared to levels measured in SARS-CoV-2 convalescent human sera (N = 66) collected at between 0.5 to 2 months post symptom onset. The same in-house ELISA was used for the macaques and human samples. (**D**) Spike-specific IFN-γ+ T cell responses measured 2 weeks after the last vaccination. Squares represent (G1, G2, G3, G5) samples collected 2 weeks after the 3rd vaccination; triangles (G4, G6) denote samples collected 2 weeks after the 2nd vaccination. Spike-specific cellular memory responses are shown as % of memory CD4+ (left panel) and as % of memory CD8+ (right panel) T cell subset. Median values are indicated. (**E**) The fraction of Spike-specific cytotoxic (Grz+ and/or CD107+) IFN-γ+ CD8+ T cells from the samples shown in panel D.

immunization of DNA and protein (G3), comprising simultaneous administration of both vaccine components in the same anatomical site, was tested in the 3-dose vaccination protocol. Animals in G1-G4 were subjected to a pathogenic SARS-CoV-2 challenge (strain-2019-nCoV/ USA-WA1/2020) at 5 weeks after the last vaccination.

Humoral immune responses were measured over time to Spike-RBD by in-house ELISA assay (Fig 1B). Similar data were obtained measuring responses to complete trimeric Spike by ELISA and the two assays showed excellent correlation (S2 Fig). We found strong anti-Spike antibody responses in all groups with an increase of antibody titers after each vaccination, except G6. The animals in this group were vaccinated with the extended constrained S-RBD and showed poor humoral responses in macaques (Fig 1B) and in mice (S1 Fig). The G1 vaccine S-1273 which produced both soluble S1 and lower levels of the prefusion-stabilized Spike (S1A Fig) showed slightly lower immunogenicity, driven by 2 of the 4 animals with lower responses, but this did not reach significance. Otherwise, the responses induced by the DNA-only vaccine groups (G1, G2, G4, G5) in macaques were similar, although in mice responses to G5 vaccine were lower (S1C Fig).

Importantly, the responses in the DNA+Protein co-immunization group (G3) differed from each of the G1 and G2 at the p<0.0001 level (repeated measures analysis of variance), while the other two groups showed only a small difference. G3 showed the highest responses throughout the study, starting as early as 2 weeks after the 1st vaccination, at which time a 100% response rate was observed. In contrast, the DNA-only groups (G1, G2, G4, G5) showed an antibody response rate of 30% (6 of 20 animals) at week 2, which further increased to 70% (14 of 20 animals) by week 3. The antibody levels in the DNA+Protein co-immunization group continued to be significantly higher, reaching a ~7-fold higher level after the last vaccination based on a linear antibody endpoint titer comparison. These findings are in agreement with our previous data that evaluated DNA+Protein co-immunization regimens for HIV/SIV [28–31] and showed rapid induction of the highest antibody levels.

Animals in groups G2 and G4 received the same vaccine, however, G4 received only 2 vaccinations and had a longer 5-week rest period between the 1st and the 2nd vaccination compared to 3 vaccinations separated by a 3-week rest for all other groups. The anti-Spike-RBD antibody levels in G4 reached a median AUC titer of 5.9 log (range 3.2–7.9) with 3 of the 4 animals showing high level responses, including one approaching those of the DNA+Protein co-immunization group (G3) after the 2nd vaccination (median log AUC 8.5; range 7.3–8.8). Of note, for most animals the anti-Spike-RBD antibody responses induced by the priming vaccination did not peak at week 2 but were higher by week 3 in the G3-G5 groups and even higher at week 5 in 2 of 4 animals in the G4 group (Fig 1B). These data suggest that a longer interval between vaccinations (5 vs 3 weeks) may be beneficial, at least for macaque studies.

Next, using the same in-house ELISA, we found that the Spike-RBD antibody levels in vaccinated macaques were significantly higher than those found in convalescent plasma from a sub-cohort of COVID-19 patients collected at 0.5–2 months post symptom onset [40,41] (Fig 1C). The anti-Spike antibodies induced by the DNA+Protein co-immunization regimen were highest.

Analysis of antigen-specific T cell responses showed robust induction of Spike-specific IFN-$\gamma^+$ CD4$^+$ and CD8$^+$ memory T cell responses by all regimens (Fig 1D and S3 Fig). No significant difference was found among the groups which received the full-length or close to full-length immunogens after the 2nd vaccination (S3 Fig) or after the last vaccination, with some skewing towards CD4$^+$ T cell responses, except G5 which showed a stronger CD8$^+$ T cell response (Fig 1D). Even group G6 which received the extended S-RBD vaccine and failed to induce appreciable humoral responses developed T cell responses. There was greater animal-to-animal variation in cellular responses, compared to humoral responses, as noted in our

previous studies in outbred macaques [28,30–33]. The vaccine-induced T cell responses displayed robust features of cytotoxic T cells (granzyme B content and degranulation) (Fig 1E), a hallmark of the DNA vaccine platform.

Thus, the DNA-based vaccines were able to induce robust Spike-specific humoral and cellular responses including cytotoxic T cell responses. The highest antibody levels were detected in animals that received the DNA+Protein co-immunization regimen.

## Effector function of Spike DNA vaccine induced antibodies

Vaccine-induced antibodies were evaluated for their ability to neutralize the Spike pseudo-typed Nanoluc reporter virus (Fig 2A). All immunized animals (except G6, which was not tested) developed robust NAb after the 2nd and the 3rd vaccination, with macaques in the DNA +Protein group having the highest NAb levels (G3; median reciprocal ID50 titer 4 log), reflecting the higher Ab levels elicited by this vaccine regimen (Fig 1B). Vaccinated macaques had NAb levels comparable to those found in the convalescent patient sera described above (Fig 1B, right panel). Again, the NAb induced by the DNA+Protein co-immunization vaccine (G3) matched the highest NAb levels found in convalescent patients.

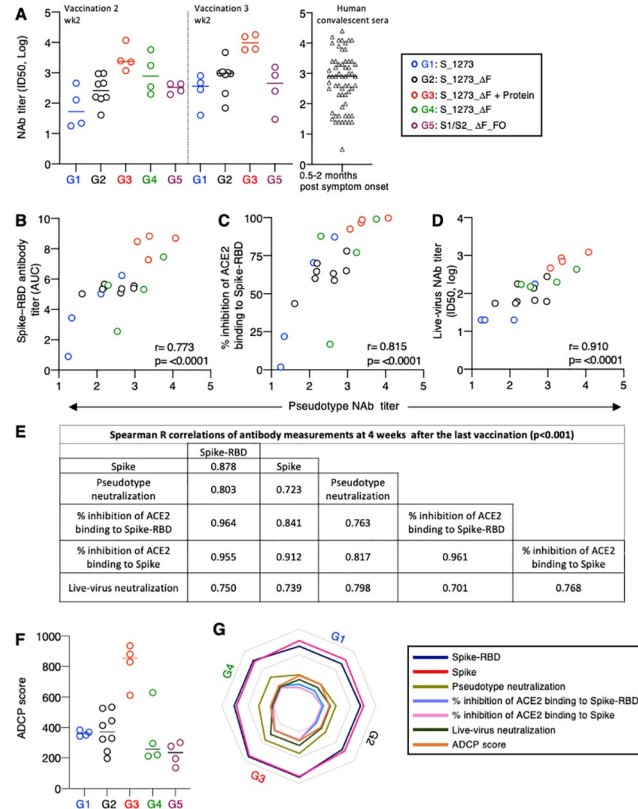

**Fig 2. Spike-antibody effector functions.** Neutralization assays were performed using Spike-1254 (Wuhan-Hu-1) pseudotyped reporter virus in HEK293/ACE2 wt cells. (**A**) NAb titers (ID50) measured in the different vaccine groups at 2 weeks after the 2nd and the 3rd vaccination were compared to NAb titers obtained in SARS-CoV-2 convalescent sera (N = 48) collected 0.5 to 2 months post symptom onset. The same in-house pseudotype NAb assay was used. Median values are indicated. (**B-D**) Spearman correlations of reciprocal pseudotype NAb titers (ID50, log) and (**B**) Spike-RBD antibody titers (ELISA, AUC); (**C**) % inhibition of ACE2 binding to Spike-RBD; (**D**) reciprocal live-virus neutralization ID50 (PRNT, log) using sera collected 2 weeks after the last vaccination. (**E**) Summary of Spearman correlations of antibody measurements 4 weeks after the last vaccination determined at one week before challenge. (**F**) Antibody-dependent cellular phagocytosis (ADCP) score is shown for the 4 vaccine groups at 4 weeks after the last vaccination. (**G**) Radar plot summarizing the serologic features.

Univariate analysis of pseudotype NAb titers (G1 to G4) and a series of functional antibody features was performed at 2 weeks after the 2nd vaccination (Fig 2B–2D) and 4 weeks after 3rd vaccination (one week before infection) (Fig 2E). A strong direct correlation was found between pseudotype NAb titers and Spike-RBD antibody levels (Fig 2B), the ability of the vaccine-induced antibodies to effectively inhibit ACE2 binding to Spike-RBD (Fig 2C), and neutralization of SARS-CoV-2 live-virus infection of Vero E6 cells (Fig 2D). These correlations persisted at 4 weeks after the last vaccination (one week before the challenge) (Fig 2E).

We also determined antibody-dependent monocyte cellular phagocytosis (ADCP) as another effector function (Fig 2F). Robust ADCP was detected in all vaccinees before challenge, with the DNA+Protein group (G3) showing significantly higher ADCP function than those induced by the DNA-only groups (ANOVA, p<0.001).

Together, these *in vitro* assays demonstrate that Spike-encoding DNA vaccine regimens induced NAb with the potential to control virus infection and that the DNA+Protein vaccine group mounted the highest responses among the different effector features tested (Fig 2G).

## Cross-reactivity of vaccine-induced binding and neutralizing Ab

The breadth of the vaccine-induced neutralizing antibodies was evaluated using the panel of Spike variants (https://www.gisaid.org/hcov19-variants/) shown in Fig 3A. The Spike proteins of variants Alpha (B.1.1.7; UK), Beta (B.1.351; ZA) and Gamma (P.1; BR) share the N501Y mutation located in RBD. Beta and Gamma also have the K417N and E484K mutations. Delta (B.1.617.2) has the L452R and L478K mutations. The Epsilon (B.1.427/B.1.429; CAL.20C; US) and Eta (B.1.525; NG) variants have a single mutation each in RBD (L452R and E484K, respectively), whereas A.23.1 (Uganda, UG) has the V367F mutation in RBD. All Spike variants have the D614G or the nearby Q613H mutation located in the C-terminal S1 region (CTD). Of note, these Spike proteins also differ in their N-terminal sequences (NTD), which is thought to also contribute to the Spike RBD-ACE2 interaction. Some of these mutations have been associated with increased virus transmissibility [42–44]. Whereas the Wuhan-HA-1 Spike induced immune responses recognized Alpha and Delta Spike-RBD similarly, binding to Beta Spike-RBD was significantly reduced (Fig 3B).

The ability of NAb induced by the DNA-only (G2) and the DNA+Protein co-immunization (G3) vaccine to neutralize reporter viruses pseudotyped with different Spike variants was tested (Fig 3C and S4 Fig). The same gradient of neutralization efficacy was observed in the DNA-only (Fig 3C, left panel) and DNA+Protein (Fig 3C, right panel) vaccinated macaques, with Wuhan-Hu-1 being the most susceptible, similar to Alpha, Delta, Epsilon, Eta and A.23.1, followed by lower neutralization of Beta and Gamma Spike pseudotyped viruses. Both vaccine groups showed greatly reduced neutralization of Beta and Gamma (Fig 3C), reflecting their overall lower levels of humoral responses recognizing Beta Spike-RBD (Fig 3B). Together, these data point to K417N to play a key role in the impaired binding and neutralization of this Spike variant by Wuhan-HA-1 induced antibodies. In general, the DNA+Protein vaccine induced substantially increased neutralization compared to DNA-only vaccine. The cross-neutralization breadth of the DNA+Protein vaccine was reduced by ~11x (Gamma) to ~22x (Beta) compared to Wuhan-Hu-1, similar as a previous report [45]. Both vaccine groups showed robust NAb responses to Alpha and Delta Spike carrying pseudotype viruses (Fig 3C). Sera from the DNA+Protein group also neutralized Spike variants Epsilon, Eta and A23.1 (Fig 3C, right panel and S4 Fig). No significant difference was found between the inhibition of reporter viruses carrying the Wuhan-Hu-1 Spike and Epsilon Spike, the latter of which has a single L452R mutation in RBD. A weaker but not significant inhibition (~2x) was observed with the

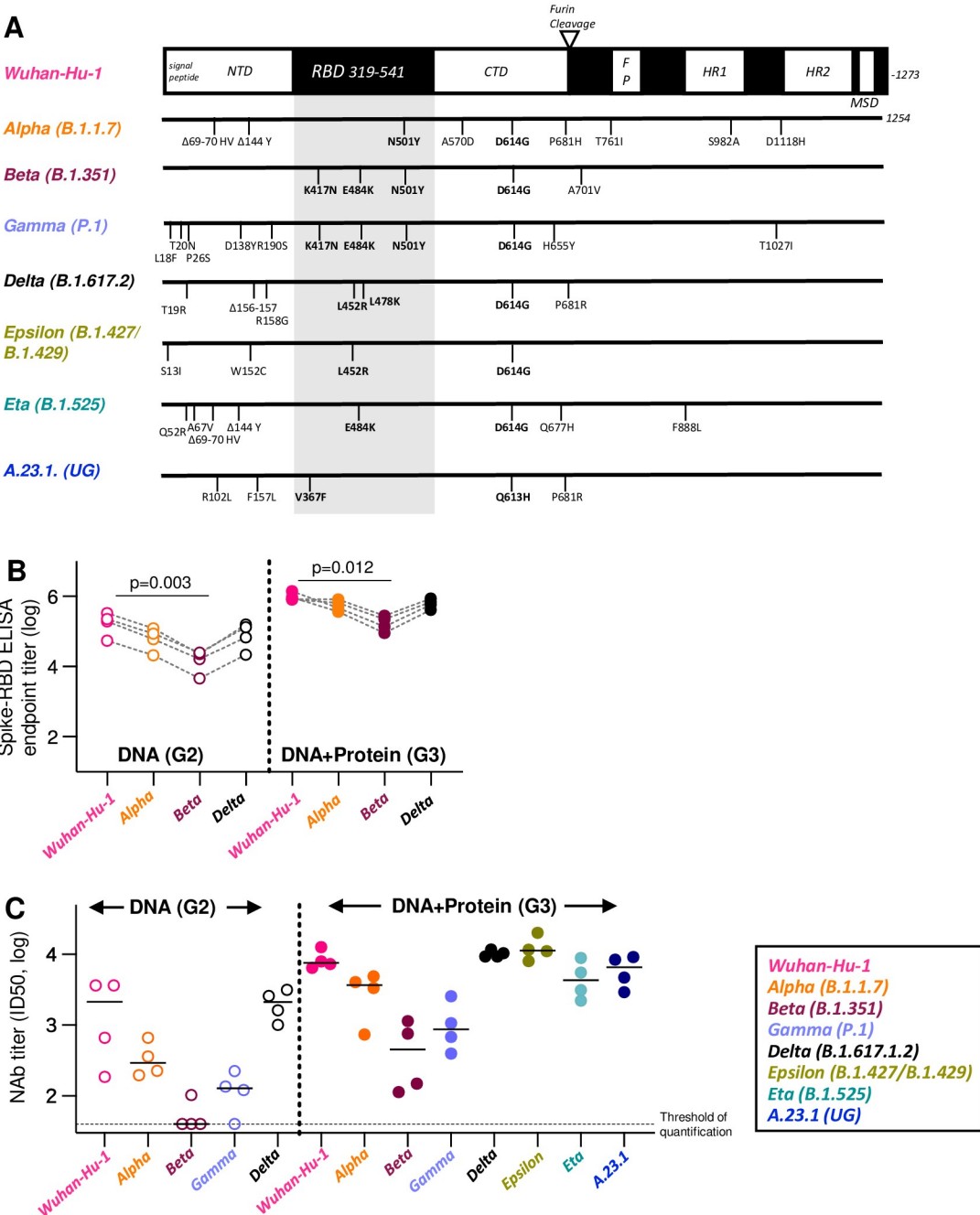

**Fig 3. Induction of cross-reactive binding and neutralizing antibodies.** (**A**) Schematic representation of a panel of Spike variant proteins used in pseudotype virus neutralization assay. RBD is indicated in the grey shaded area. All Spike variants, have the D614G change. (**B**) Cross-binding to Spike-RBD of sera collected 2 weeks after the last vaccination of the group that received DNA only (G2; left panel; N = 4) compared to DNA+Protein group (G3; right panel; N = 4) are shown for Wuhan-Hu-1, Alpha, Beta and Delta Spike-RBD. P values are from ANOVA, Friedman test. (**C**) Cross-neutralization of sera collected 2 weeks after the last vaccination (same sera as in panel B) of the group that received DNA only (G2; left panel) compared to DNA+Protein group (G3; right panel) are shown for Wuhan-Hu-1 Spike and the indicated variant Spike proteins. Reciprocal pseudotype NAb ID50 titers (log) are plotted for individual animals. The NAb ID50 threshold of quantification (0.5 log) and the threshold of detection (0.1 log) in this assay are indicated. Median values are indicated.

A23.1 and Eta Spike carrying reporter viruses, compared to that of Alpha. These proteins have a single AA mutation in RBD in addition to the Q613H/D614G mutation.

Together, DNA-only or DNA+Protein vaccine regimens induced Wuhan-Hu-1 Spike NAb in macaques with potent and broad *in vitro* cross-neutralization capabilities. Importantly, although the neutralization breadth (including Alpha, Beta and Gamma) was similar to SARS-CoV-2 convalescent plasma or induced by the SARS-CoV-2 mRNA vaccine reported previously [46–50], the current study revealed rank-ordered neutralization capability against a series of additional Spike variants with distinct mutations in RBD.

## Control of CoV-2 challenge in macaques

Five weeks after the last vaccination, groups G1 to G4 and 4 naïve control macaques were challenged with SARS-CoV-2 (strain-2019-nCoV/USA-WA1/2020) via a combined intranasal (IN) and intratracheal (IT) route. Virus loads (VL) were measured using the subgenomic (sg) Nucleocapsid (N-sg) (Fig 4A) and Envelope (E-sg) (Fig 4B) PCR assays. VL were measured at days 1 and 3 in nasal and in pharyngeal swabs, respectively, and on day 4 in bronchoalveolar lavage (BAL) (Fig 4A and 4B). Although both assays measured viral mRNA transcripts the N-sg assay was more sensitive.

All control animals became infected following challenge, with positive VL at day 1 and increasing VL by day 3 in the nasal swabs (Fig 4A and 4B, **left panels)**, positive VL in pharyngeal swabs at day 1 and 3 (Fig 4A and 4B, **middle panels)** as well as in the lungs (BAL) at day 4 (Fig 4A and 4B, **right panels)**. All control animals also developed anti-Spike antibodies by 8–10 days post-infection (Fig 5A).

In contrast to the controls, all vaccinated animals showed control of virus replication and lack of dissemination to the lower respiratory tract. Indeed, all animals in DNA+Protein vaccine group (G3) had significantly lower or undetectable VL, in all assays, tissues and sampling times (Fig 4A and 4B, **left panels)**. On the other hand, in the DNA-only vaccine groups (G1, G2, G4), virus replication was detected on day 1 in nasal swabs by the N-sg RNA VL assay. In contrast to the control group, however, by day 3 viral loads were significantly reduced in the G1 and G2 groups. Interestingly, similar as in the control group, the G4 (2x DNA vaccinations) had high viral loads in nasal swabs at day 1 that diminished only marginally by day 3. However, in contrast to the control group, no viral RNA was detected at day 4 in the lung (BAL) in none of these 3 vaccine groups, and in was only infrequently detected in the pharyngeal mucosa at days 1 or 3 (Fig 4A and 4B, **middle and right panels)**. Hence, the DNA only vaccines elicited robust control of infection, especially in the lower airways. Regardless of regimen, the DNA vaccines afforded robust protection against SARS-CoV-2 infection in the lung, whereas the DNA+Protein vaccine provided the best protection in the upper respiratory tract.

To understand the contribution of vaccine-induced antibody responses to infection control, univariate correlations were performed using the set of humoral immune response data obtained at 2 weeks after the last vaccination (corresponding to 3 weeks before challenge) (Fig 4C–4E) and one week before challenge (Fig 4F–4I). Strong inverse correlations with virus loads at day 1 in nasal swab were noted with (i) Spike-RBD antibody titers (r = -0.452, p = 0.04; Fig 4C; r = -0.623, p = 0.003; Fig 4F); (ii) pseudotype NAb levels (r = -0.480, p = 0.032; Fig 4D); (iii) inhibition of ACE2 binding to Spike-RBD (-0.638, p = 0.002; Fig 4G); (iv) live-virus neutralization titer (r = -0.46, p = 0.04 r = -0.547, p = 0.012; Fig 4E; r = -0.547, p = 0.012; Fig 4H); and (v) antibody-dependent cellular phagocytosis (ADCP) (r = -0.563, p = 0.009; Fig 4I). Similar strong inverse correlations were found with antibody measurements and virus loads at both time points. These data show continuing vaccine-induced humoral response able to control of SARS-CoV-2 replication. In contrast, the pre-challenge vaccine-

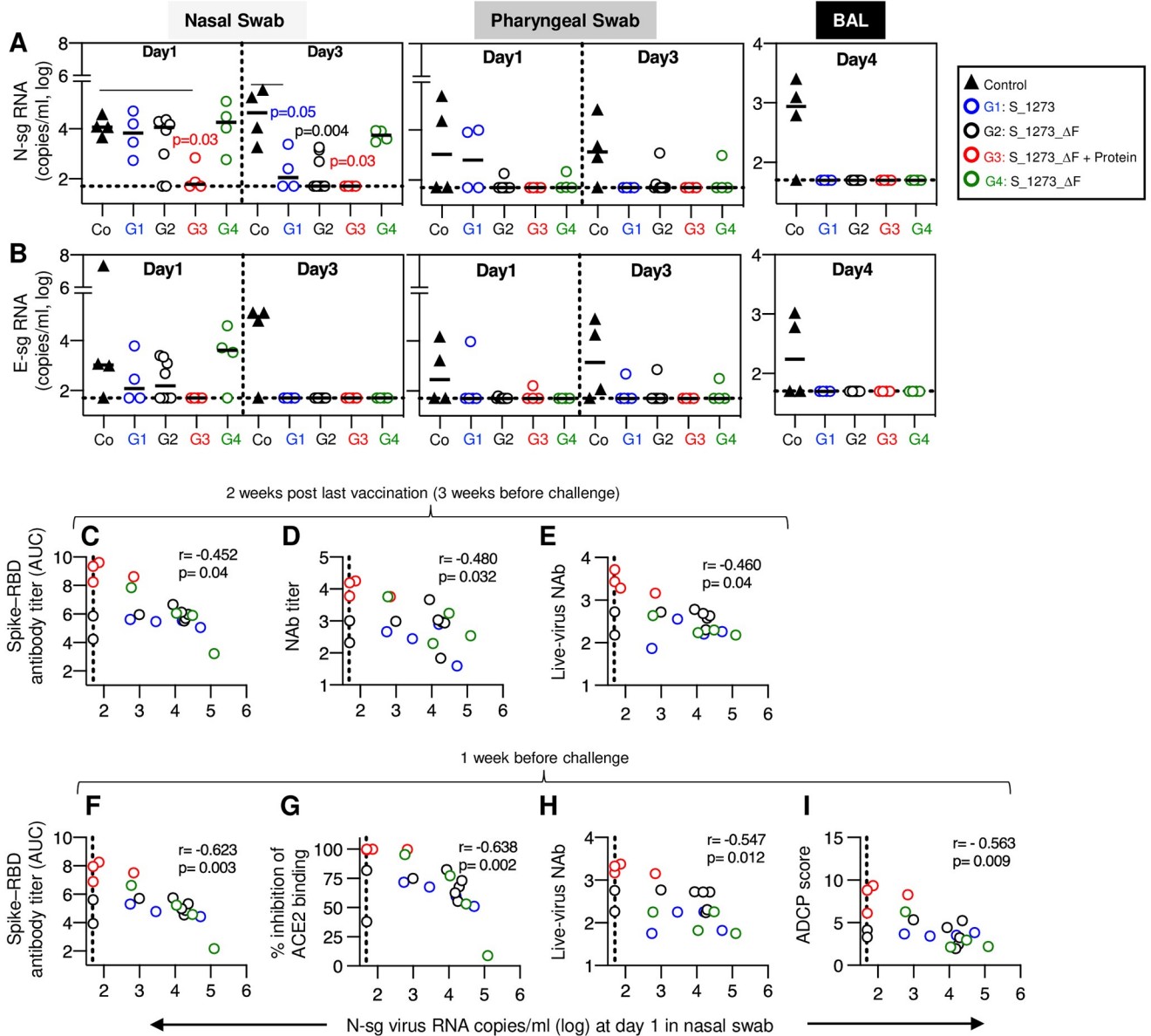

**Fig 4. Virus loads and immune response in rhesus macaques challenged with infectious SARS-CoV-2 virus.** (**A, B**) Virus load measurements upon reverse transcription polymerase chain reaction (RT-PCR) targeting (**A**) N-subgenomic (sg) RNA and (**B**) E-sg RNA isolated from nasal swabs (left panels) and pharyngeal swabs (middle panels) collected at day 1 and day 3 post-challenge and from bronchoalveolar lavage (BAL, right panels) collected at day 4 are shown. The p values (color-coded) denote significance of virus loads comparing vaccinees to controls analyzed at day 1 and 3 postinfection in nasal swab using the N-sg assay. Median values are indicated. (**C-I**) Univariate correlations of virus loads in nasal swabs on day 1 post challenge and antibody measurements in samples collected (C-E) 2 weeks after the last vaccination (representing 3 weeks before challenge) and (F-I) one week prior to challenge are shown with (**C, F**) Spike-RBD ELISA antibody titers (AUC); (**D**) reciprocal pseudotype NAb ID50 titer (log); (**E, H**) reciprocal live-virus NAb titer (ID50, log); (**G**) % inhibition of ACE2 binding to Spike-RBD; (**I**) ADCP score. Dotted line denotes threshold of the assay (<50 copies/ml).

induced T cell responses did not correlate with virus control, likely due to the strong contribution of the humoral immune responses and the very short course of infection (few days) in this model.

Together, anti-Spike antibody levels, neutralization ability and ability to mediate ADCP effector function were predictive of challenge outcomes. Thus, DNA-based vaccines, either

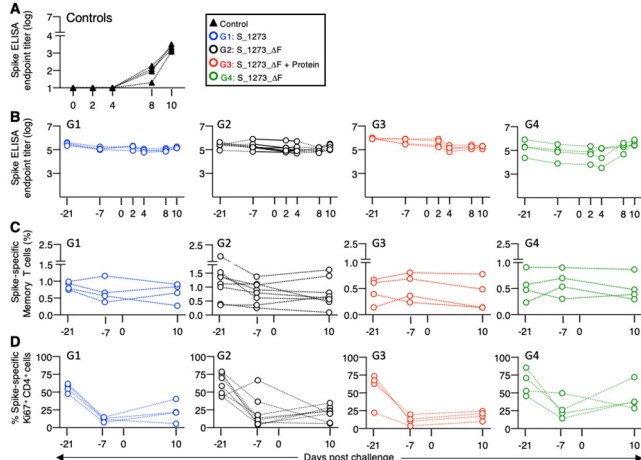

**Fig 5. Post-challenge immune responses.** (**A, B**) Anti-Spike antibodies measured by ELISA (reciprocal endpoint titer, log) in the (**A**) control animals over the course of infection up to day 10, and (**B**) in the vaccinees at -day 21 (2 weeks after the last vaccination) and -day 7 (4 weeks after the last vaccination) before challenge and days 2, 4, 8, and 10 after challenge. (**C, D**) Antigen-specific memory T cell responses were monitored from 2 weeks after the last vaccination to 10 days after infection showing (**C**) % of Spike-specific total IFN-γ$^+$ memory T cells and (**D**) Spike-specific Ki67$^+$ IFN-γ$^+$ memory CD4$^+$ T cells, shown as % of the antigen-specific total memory CD4$^+$ T cell subset.

alone or as part of the protein co-immunization regimen were able to control SARS-CoV-2 infection in the rhesus macaque model.

## Post-challenge immune response analysis

Analysis of post challenge anti-Spike immune responses showed, as noted above, that anti-Spike antibodies were present by day 10 post challenge in all control animals (Fig 5A). In contrast, no anamnestic anti-Spike antibody response were found in vaccine groups G1-G3 (Fig 5B). Only two macaques in G4, which also had the highest and most persistent VL in nasal swabs among the vaccinees (up to day 3; see Fig 4A), developed an anamnestic anti-Spike antibody response by day 8–10 (Fig 5B).

Spike-specific T cell responses were also monitored 2 and 4 weeks after the last vaccination (challenge days -21, and -7) and post-challenge (day 10) (Fig 5C and 5D). The size of the Spike-specific total IFN-γ$^+$ memory T cell population did not significantly change in the vaccinees over time (Fig 5C). Analysis of the proliferative capacity of Spike-specific CD4$^+$ memory T cells (Ki67 expression) showed the expected contraction after the last vaccination (Fig 5D), followed by very limited proliferation after challenge that was more pronounced in G4. Thus, the anamnestic humoral and cellular responses to virus challenge in the two animals in G4 is likely associated with the slightly longer persistence of virus in the upper respiratory tract. The size of the CD8$^+$ T cell population did not change upon challenge within the timeframe measured (day 10) and T cell responses could not be correlated with virus loads (day 1). Spike-specific T cell responses were not detected in the control group during the 10-day post-infection monitoring period.

Together, our data showed induction of robust antibody and T cell responses by the DNA-only and the DNA+Protein vaccines regimens able to effectively mediate protection and to control SARS-CoV-2 infection in the rhesus macaque model. The best vaccination efficacy was achieved in animals that received the DNA+Protein co-immunization regimen.

## Discussion

In this report, two types of SARS-CoV-2 vaccination protocols were explored, DNA-only and simultaneous co-immunization of DNA and Protein at the same anatomical site. Immunogenicity was evaluated for DNA vaccines expressing prefusion stabilized Spike proteins differing in the presence or absence of the furin cleavage site and C-terminal region in mice and rhesus macaques. The DNA+Protein regimen combined Spike-ΔF to maximize the prefusion structure together with soluble GLA-SE adjuvanted Spike-RBD protein. Each vaccine regimens induced antibodies able to neutralize SARS-CoV-2 *in vitro* and *in vivo* and mounted robust protection against SARS-CoV-2 replication in the lower respiratory tract of the macaques. Importantly, DNA+Protein co-immunization regimen induced the highest magnitude of binding and neutralizing antibodies and afforded an improved virologic control in the upper respiratory tract.

The results of SARS-CoV-2 DNA+Protein co-immunization were consistent with our previous findings from HIV/SIV DNA+Protein co-immunization studies, where co-immunization in the same site provided the highest antibody responses, while also inducing T cell responses [28–33]. Here, DNA+Protein immunization induced the most robust anti-Spike antibody responses, evident after a single vaccination, and these levels remained higher throughout the study, also consistent with previously reported in HIV/SIV vaccine studies [28–33]. During the completion of our report, Li et al [51] reported a vaccine regimen in which DNA and protein were administered simultaneously (but in separate anatomical sites). They found better protection in this group compared to DNA or protein only vaccines. These data are in overall agreement with our current study and support the inclusion of both vaccine components starting with the priming vaccination. Importantly, however, we have previously shown that administration of HIV-Env DNA and protein in the same sites resulted in significant differences in antibody magnitude quality, and superior protective immunity from SHIV infection versus DNA and protein administered in contralateral sites [28]. We hypothesize that simultaneous recognition, processing, and presentation of DNA+Protein in the same draining lymph nodes contribute to the development of optimal immunity.

Protection from SARS-CoV-2 infection depends on the presence of vaccine-induced immune responses in mucosal tissues. We previously showed that HIV/SIV DNA vaccines administered intramuscularly (IM) induced potent systemic immune responses that efficiently disseminate to mucosal sites [31,33,52,53]. Here we showed that IM-delivered Spike DNA induced humoral responses able to control against combination intranasal/intratracheal SARS-CoV-2 challenge. The concept of IM-delivery of DNA to induce immune responses with mucosal dissemination has also been demonstrated in clinical trials using an HPV DNA vaccine [54,55].

We noted a difference in the extent of protective efficacy between vaccine groups which received 2 or 3 Spike DNA vaccinations. Animals in the 2x vaccination group (G4) showed the highest, although transient, local viremia without dissemination through the respiratory tract, despite similar magnitude and function of antibodies measured in the blood. These data suggest a possible difference in mucosal immune responses between the two regimens, resulting in diminished efficacy in the early control of the infection after 2 immunizations. Three-dose DNA+Protein co-immunization induced the best protective responses with rapid control of the initial SARS-CoV-2 infection. Antibody levels, their neutralizing capabilities and their ADCP effector functions correlated with control of viremia and virus dissemination, and these responses were greatly increased in the DNA+Protein co-immunized animals. Greater protection by NAb with effector function induced by Spike-DNA has been reported in macaques [21]. These data are in agreement with our findings. Our optimized Spike DNA+Protein vaccine regimen maximizes these features.

Despite the difference in protection between the 2x and 3x DNA vaccination regimens, an intriguing difference in humoral immune responses could be attributed to the different rest periods (5 weeks versus 3 weeks). At least in macaques, and possibly also in humans, a more prolonged rest time between vaccinations may be advantageous. The vaccination schedule for the EUA approved SARS-CoV-2 mRNA vaccines, with rest periods of 3 and 4 weeks, may also be improved by lengthening the rest period, thereby benefitting public health needs.

An important feature of the SARS-CoV-2 DNA vaccine is its ability to induce robust cellular as well as humoral immune responses (for reviews see [14–20]). In contrast to mRNA vaccines that induce robust antibody responses and low T cell responses, several clinical trials have shown that IM- as well as intradermal (ID)-delivered DNA vaccines [12,56–60] are able to induce both humoral and cellular responses and that the latter is a critical component to maintain durable antibody responses exerted by CD4$^+$ T help. Although the underlying mechanism explaining the difference in immunity induced by DNA versus mRNA vaccines is not clear, the ability to induce T cell responses is important not only for antibody development but also as a second line of defense in protection from spreading infection and disease development. It is possible that a 3$^{rd}$ booster mRNA vaccination may alleviate this. Our overall experience with DNA vaccines has been that they provide superior cellular immunity in numbers and duration, and the hypothesis to test is that it may be of benefit for the establishment of long-term protective immunity. On the other hand, delivery of DNA via the ID and IM routes in clinical trials has shown the potency of the vaccine platform [12,56–60]. The combination of DNA with protein delivered simultaneously via the IM route in the same anatomical site further advances the nucleic acid vaccine platform and increases the simultaneous induction of maximal humoral and cellular immune responses.

Due to the differences in vaccine regimens, challenge virus stocks, animal facilities, methods to monitor immune responses, and virus load measurements used, it is challenging to make a comparison among the reported NHP studies. The overall success of several vaccine regimens in preclinical models and the rapid advancement of clinical trials and the EUA in the US and Conditional Marketing Authorization in the European Union, of several vaccines in less than one year from recognition of the SARS-CoV-2 outbreak is a true triumph of modern medicine (reviewed in [1,2]). SARS-CoV-2 Spike protein has shown to be a potent immunogen and has been sufficient in several vaccine platforms to induce protective immunity. Although the durability of the protective immunity is unclear, induction of a balanced immune response, including cellular immune responses, afforded by DNA vaccination may extend this immunity.

## Material and methods

### Ethics statement

The mouse studies were approved by the National Cancer Institute-Frederick Animal Care and Use Committee and were conducted in accordance with the ACUC guidelines and the NIH Guide for the Care and Use of Laboratory Animals (Approval ASP#20–024). The nonhuman primate studies were conducted in compliance with all relevant local, state, and federal regulations and were approved by the BIOQUAL Institutional Animal Care and Use Committee (IACUC) (Approval #20–042). Human serum samples were used from donors who participated in an ongoing phase 2 study (NCT04408209). Donors gave written informed consent. Research approved by the NCI at Frederick Institutional Biosafety Committee (Approval IBC#19–53 and #2020–13).

## SARS-COV2 DNA vaccine

All plasmids contain RNA/codon-optimized genes inserted between the human CMV promoter and the BGH polyadenylation signal of the expression vector pCMV.kan [61]. Endotoxin-free DNAs (cat#12391; Qiagen, Valencia, CA) were prepared according to the manufacturer's protocol. RNA-optimized Spike genes were designed based on the SARS-CoV-2 Spike protein sequence from Wuhan-Hu-1 (GenBank: MN_908947) with its original signal peptide and the DNAs were synthesized by GeneArt (Thermo Fisher Scientific). All Spike plasmids have the 2P stabilizing mutations by substituting of $K_{986}V_{987}$ [34–36]. The Spike_DF proteins have a furin cleavage site mutation by replacing the residues $_{679}$NSPRRA$_{684}$ with residues acids I and L [37]. S_1273 (plasmid C57) and S_1273_ΔF (plasmid C58) express the full-length 1273 amino acid Spike. S_1208_FO (plasmid C12) has a deletion of the furin cleavage site and has the C terminus (AA 1209–1273) is been replaced by trimeric foldon of the T4 bacteriophage [62]. Plasmid S-RBD_ΔF (plasmid C36) produces a protein comprising the Spike CoV-2 signal (AA1-14), Spike AA 295–1171 and the T4 foldon. Two additional mutations were made K304E and S944E to allow for putative electrostatic interactions with K964 and K310, respectively.

## *In vitro* analysis of the Spike DNA plasmids

Expression of Spike DNA plasmids was analyzed upon transient transfection of HEK293T clone17 cells using the Calcium Phosphate DNA co-precipitation and DNA (1 microG DNA/ 60 mm plate) as described [32,63]. After 48 hours, supernatants and cells were harvested, and the cells were lysed in 1 ml of hypertonic N1 lysis buffer as described. Protein production was measured by Western immune-blotting by loading same amount of cell lysate or supernatant on Novex Bolt 8% Bis-tris (Thermo Fischer Scientific) and blotted onto nitrocellulose membranes which were probed with the SARS-CoV2/2019-nCoV Spike rabbit polyclonal Ab (PAb, cat# 40591-T62, Sino Biologicals; dilution 1:1,000) followed by anti-Rabbit IgG-HRP conjugated antibody (1:15,000 dilution, GE Healthcare, Piscataway, NJ). The protein bands were detected using the enhanced chemiluminescence (ECL) Prime Western blotting detection system (GE HealthCare, Piscataway, NJ). The images of the blot were acquired on the ChemiDoc XRS using Bio-Rad ImageLab software (Bio-Rad laboratories, Inc).

## Vaccination of mice

Female BALB/c 6 weeks-old were obtained from Charles River Laboratory (Frederick, MD). Five mice per group were vaccinated by the *in vivo* electroporation method (AgilePulseTM System; BTX, Harvard Bioscience, Inc) at weeks 0 and 3. Endotoxic-free DNA (25 microG DNA in 25 microL) was injected intramuscularly in the left quadriceps at week 0 and in right quadriceps at week 3. Two weeks later, the mice were sacrificed.

## Vaccination and challenge of rhesus macaques

For the immunogenicity study, animals (N = 4) were vaccinated at week 0 and 3 with 2 mg of S1/S2_ΔF_FO (plasmid C12) and 2 mg of S-RBD_ΔF_FO (plasmid C36), respectively. For the challenge study, 3 groups (G1, G2, G3) were vaccinated three times at week 0, 3 and 13 and one group (G4) was vaccinated twice at week 0 and 5. G1 received 2 mg of S_1273 (plasmid C57) and G2 and G4 received 2 mg S_1273_ΔF (plasmid C58). All vaccine mixtures contained in 0.2 mg rhesus IL-12 (Plasmid AG157). G3 received 2 mg S_1273_ΔF together with 100 microG of Spike-RBD adjuvated with TLR-4 agonist (GLA-SE) (Infectious Disease Research Institute Seattle, WA). The Spike-RBD protein, spanning AA 319–525, was purified from

mammalian Expi293-cells. The protein was injected in the same anatomical site after the DNA injection. DNA vaccination was performed by intramuscular injection followed by electroporation (IM/EP) with CELLECTRA 5P (Inovio Pharmaceuticals Inc.). Five weeks after the last vaccination, the animals (G1, G2, G3, G4) together with 4 naïve controls were challenged with 10^5 PFU of SARS-CoV-2 (strain-2019-nCoV/USA-WA1/2020, obtained from BEI Resources, cat no. NR-53780, Lot# 70038893, 1.99 x 10^6 TCID50/mL) administered via the combination intranasal and intratracheal route (1 mL (0.5 mL/nostril) and 1 mL IT per animal).

## Anti-Spike antibodies ELISA

An in-house ELISA assay described elsewhere [40,41] was used to detect the Wuhan-HA-1 Spike receptor binding domain (Spike-RBD) (AA 319–525) and the complete trimeric Spike (amino acid (AA) 15–1208_2P) using mammalian Expi293-cells produced proteins [64]. A panel of HIS-tagged Spike-RBD variants was included including Alpha (N501Y), Beta (K417N/E484K/N501Y) and Delta (L452R/T478K). Briefly, the levels of antibody were measured by 4-fold serial dilutions starting at 1:50. The threshold of the Spike-RBD assay was calculated using the serum samples from 12 naïve rhesus macaques and 9 naïve BALB/c mice, respectively using the 1:200 and 1:819,200 dilutions. The cut-off values were defined using mean plus 2 standard deviations (STD) (macaque) and mean plus 1 STD (mouse) measured from OD values obtained between the 1:200 to 1:819,200 dilutions. Antibody levels were expressed as area-under-the curve (AUC) values covering serial serum dilutions (log10 transformed) above the cut-off. If a sample has an endpoint titer of 1:50, it is considered below threshold of the assay and the value is entered as 0.1. A model fit approach was conducted in R to model the curve to more accurately define endpoint titers [41]. Comparisons of antibody levels were performed using linear endpoint titer values.

## Spike SARS-CoV-2 neutralization assay

The Spike pseudotyped $pHIV_{NL}\Delta Env$-Nanoluc assay [39,65] was performed as previously described [40]. Different Spike variants including Wuhan-Hu-1, Alpha (B.1.1.7; UK), Beta (B.1.351; SA), Gamma (P.1; BR), Delta (B.1.617.2); Epsilon (B.1.429/B.1.429; US), Eta (B.1.525) and A.23.1 (Uganda, UG) spanning Spike AA 1–1258 were cloned as RNA-optimized genes into CMV.kan. Briefly, pseudotyped viruses carrying different Spike proteins were generated in HEK293T/clone17 cells by co-transfection with $pHIV_{NL}\Delta Env$-Nanoluc. The pseudotyped viruses were titrated by serial dilution (starting at 1:5 followed by six 10-fold dilutions). The undiluted and the 7 dilutions (100 microL; $1x10^5$ Luminescence (RLU)) were used in triplicate to transduce HEK293T/ACE2wt cells. Four-fold serial dilutions of heat-inactivated sera (total of 8 dilutions: 1:40 to 1: 655,360) were assayed in triplicates in HEK293T/ACE2wt cells and luciferase levels in the cell lysates were measured. The 50% Inhibitory dose (ID50) was calculated using GraphPad Prism version 9.0.2 for MacOS X (GraphPad Software, Inc, La Jolla, CA) with nonlinear regression curve fit using inhibitor vs responses variable slope (four parameters). ID50 reciprocal endpoint titers were calculated as percentage of neutralization over the levels of luciferase measured at the 1:655,360 dilution using a model fit method conducted in R of the neutralization curves. The NAb ID50 threshold of quantification is 0.5 log and the threshold of detection is 0.1 log in this assay. The MesoScale Discovery (MSD) Binding competition assay (K15386U) with SULFO-TAG–labeled ACE2 was measured at a 1:100 dilution of sera.

## Plaque reduction neutralization test (PRNT)

The PRNT assay were conducted in Vero E6 cells (ATCC, cat# CRL-1586) plated in 24-well plates at 175,000 cells/well in DMEM + 10% FBS + Gentamicin reaching 80–100% confluency the following day. Heat-inactivated serum samples were subjected to five 3-fold serial dilutions starting with a 10-fold dilution (first dilution corresponds to 1:30 of serum). Diluted sera (300 microL) were incubated with 300 microL of medium containing 30 pfu of virus and incubated for 1 hour (corresponding to 1:60 dilution of starting serum). The titrated virus-sera mixture (250 μL/well; duplicate assay) was used to replace the medium and the cells were incubated for 1 hour to allow virus infection. Then, 1 mL of the 0.5% methylcellulose media was added per well and the plates were incubated for 3 days. Then, the cells were fixed with ice cold methanol at -20˚C for 30 minutes and stained with 0.2% crystal violet in 20% MeOH for 30 minutes at room temp. The plates were washed once and let dry for ~15 min. The plaques were recorded for each well and the ID50 titers were calculated based on the average number of plaques detected in the virus-only control wells.

## Antibody-dependent cellular phagocytosis (ADCP)

The ADCP assay was performed as described [66] using purified Spike-RBD-coated yellow-green fluorescent beads. The beads (100,000/reaction) were incubated at 37˚C with a 2,500-fold dilution of sera from vaccinated macaques and 20,000 THP1 effector cells. After 3 hrs, the cells were pelleted, washed, and fixed. The level of phagocytosis was quantified as the composite of the percent and median fluorescence intensity of bead uptake. ADCP score was calculated as: % Spike-RBD-bead-positive cells × MFI of positive population)/10^4.

## Human convalescent serum samples

Serum samples were obtained from COVID-19 infected persons (NCT04408209) and represent a sub-cohort from a previously described cohort [40,41].

## SARS-CoV-2 -specific T cell responses in PBMC

Antigen-specific T cells were measured by flow cytometry as previously described [28,32,53]. Briefly, Ficoll-Paque Plus (GE-17-1440-03, GE Healthcare, Sweden) purified PBMC were cultured in 96-well plates for 12 hours in the presence of SARS-CoV-2 Spike peptide pools (Pep-Mix SARS-CoV-2 JPT or Biosynthesis in 2 pools), at a final concentration of 1 μg/ml for each individual peptide. PBMC cultured in the medium without peptide stimulation or in the presence of a commercial cell stimulation cocktail containing PMA and ionomycin (Cat#: 00–4970, eBioscience, San Diego, CA), respectively, were used as negative and positive controls, respectively. The protein transport inhibitor monensin (GolgiStop, Cat#51-2092KZ, BD Biosciences, San Jose, CA) was added to the wells to prevent cytokine secretion 60 minutes after the addition of the peptides. Spike-specific T cell responses were evaluated by combining surface and intracellular cytokine staining followed by polychromatic flow cytometry. The following antibodies were used in the cocktail for surface staining: CD3-allophycocyanin (APC)-Cy7 (clone SP34-2, Cat#557757, BD Pharmingen, San Jose, CA), CD4-PE-CF594 (clone L200, Cat#562402, BD Horizon, San Jose, CA), CD8-Brilliant Violet 650 (BV650) (clone RPA-T8; Cat#301042, BioLegend, San Diego, CA), CD28-peridinin chlorophyll protein (PerCP) Cy5.5 (clone CD28.2; Cat#302922, BioLegend, San Diego, CA), CD95-fluorescein isothiocyanate (FITC) (clone DX2; Cat#556640, BD Pharmingen, San Jose, CA) and CD107a-phycoerythrin (PE) monoclonal antibody (clone eBioH4A3, Cat#12-1079-42, eBioscience, San Diego, CA). After cell permeabilization, intracellular staining was performed using IFN-γ-phycoerythrin

(PE) Cy7 (clone B27, Cat#557643, BD Pharmingen, San Jose, CA), Granzyme B- allophycocya-nin (APC) (clone GB12, Cat#MHGB05, Invitrogen), and Ki67-Alexa Fluor-700 (clone B56, Cat#561277, BD Pharmingen, San Jose, CA) antibodies. Samples were acquired on a Fortessa or Symphony flow cytometer (BD Biosciences, San Jose, CA), and the data were analyzed using FlowJo software (Tree Star, Inc., Ashland, OR). Samples were considered positive if the frequency of cytokine-positive T cells was 2-fold higher than that of unstimulated medium-only control and greater than 0.05 after subtracting the medium control value. The sum of responses to Spike peptide pool 1 and pool 2 was reported as total Spike-specific T cell response. The memory T cell subset was identified as $CD95^+$ ($CD28^+ CD95^+$ and $CD95^+CD28^-$) within the $CD4^+$ and $CD8^+$ T cells, respectively.

## Subgenomic CoV-2 mRNA assay

Subgenomic (sg) SARS-CoV-2 RNA levels were measured using RT-PCR by BIOQUAL, Inc. as described previously [23]. Briefly, RNA was extracted from nasal and pharyngeal swabs and BAL fluid samples and reverse transcribed. The cDNAs were amplified in duplicate to quantify the viral subgenomic RNAs. For quantification of viral loads, a standard curve of Ct values was generated with a known copy number of serially diluted recombinant N or E DNA plasmids, respectively. The assays have a cutoff value of 50 copies.

N gene primers/probe sequences:
SG-N-F: CGATCTCTTGTAGATCTGTTCTC
SG-N-R: GGTGAACCAAGACGCAGTAT
FAM- TAACCAGAATGGAGAACGCAGTGGG -BHQ
E gene primers/probe sequences:
SG-F: CGATCTTGTAGATCTGTTCCTCAAACGAAC
SG-R: ATATTGCAGCAGTACGCACACACA
FAM-ACACTAGCCATCCTTACTGCGCTTCG-BHQ

## Statistical analysis

Analysis of the immunologic and challenge data was performed by using GraphPad Prism Version 9.0.2 X (GraphPad Software, Inc, La Jolla, CA). Comparison between the groups were made using ANOVA Kruskal-Wallis/Dunn's multiple comparison test. Correlations were performed using two-tailed P Spearman correlation test. For the model fit determination of the endpoint titers, the right side of the sigmoid dilution curve (all points after the largest drop in measured value or the highest four dilution points, which ever was longer) was fit to a self-starting asymptotic regression model (R functions SSasymp() and nls() from the "stats" R package) used to determine the nonlinear least-squares estimate of the model parameters (https://www.rdocumentation.org/packages/stats/versions/3.6.2/topics/nls) [67,68]. To determine the half value (y = 50) of the curves, all but the last dilution were fit to a self-starting four parameter Weibull function using the R drm() function from the "drc" library.

## Supporting information

**S1 Fig. Expression of different Spike DNA and Immunogenicity DNA vaccines in mice.** (A) Western blot analysis of transfected HEK293 cells detects the Spike proteins associated with the cell (top panel) and supernatant (bottom panel) using an S1-Spike specific antibody. Expression of S-1273 DNA resulted in the production and secretion of the cleaved S1 product while full-length Spike was found in the cell-associated fraction (lanes 1–2). All Spike proteins with the ΔF mutation (lanes 3–4 and lanes 5–6, respectively), including the S-RBD (lanes 7–8), remain mostly in the cell-associated fraction. Duplicate transfections are shown. (B-D) Mice

were vaccinated twice (week 0, 3) with the indicated DNAs. (C) Anti-Spike-RBD antibody responses were measured by ELISA and shown as AUC titer (log) established by model fit approach. Open circle and square symbols for the S_1273 and the S_1273_ΔF groups denote two independent studies which showed similar anti-Spike-RBD antibody levels, presented here as pooled data. Median values are indicated. P values are from non-parametric ANOVA (Kruskal-Wallis test). (D) Correlation of Spike-RBD ELISA titers (AUC) and reciprocal pseudotype NAb ID50 titer (log).
(TIF)

**S2 Fig. Correlation of Spike-RBD and trimeric Spike by ELISA.** ELISA measuring antibodies against S-RBD and trimeric Spike proteins in plasma collected at 2 and 4 weeks after the 3rd vaccination showing excellent correlations among the assays.
(TIF)

**S3 Fig. T cell responses after the 2nd vaccination.** Spike-specific IFN-γ+ memory T cell responses, measured 2 weeks after the 2nd vaccination of all groups are shown as % of memory CD4+ (left panel) and as % of memory CD8+ (right panel) T cell subset. The data from G4 and G5 are also shown in Fig 2D. Median values are indicated.
(TIF)

**S4 Fig. Cross-reactive neutralization of a panel of Spike variants.** The % neutralization of the DNA-only (G2) and the DNA+Protein (G3) vaccinated macaques against a panel of Spike variants are shown. Neutralization is calculated for each assay and plotted over the serial reciprocal serum dilutions. Mean and SEM are shown.
(TIF)

## Acknowledgments

We thank B. Nagy for support with the mouse studies; D. Weiss, J. Misamore, J. Treece, A. Cook, R. Brown, and staff (BIOQUAL, Inc.) for excellent support with the macaque studies; BEI Resources and the CDC as the source of the SARS-CoV-2 challenge stock, NR-53780; Y. Wang and J. Inglefield (Clinical Services Program, NCI) for technical support with the MSD assay; C. Fox, Infectious Disease Research Institute, Seattle for the GLA-SE; D. Esposito and J. Jones (Protein Expression Lab, Frederick National Laboratory for Cancer Research) for SARS-CoV-2 proteins; M. Ackerman and A.R. Crowley for advice with the ADCP assay, C. Bergamaschi and A. Valentin for discussion, and T. Jones for assistance.

The content of this publication does not necessarily reflect the views or policies of the Department of Health and Human Services, nor does mention of trade names, commercial products, or organizations imply endorsement by the U.S. Government.

## Author Contributions

**Conceptualization:** Margherita Rosati, George N. Pavlakis, Barbara K. Felber.

**Data curation:** Margherita Rosati, Barbara K. Felber.

**Formal analysis:** Margherita Rosati, George N. Pavlakis, Barbara K. Felber.

**Funding acquisition:** George N. Pavlakis, Barbara K. Felber.

**Investigation:** Mahesh Agarwal, Xintao Hu, Santhi Devasundaram, Bhabadeb Chowdhury, Jenifer Bear, Robert Burns, Laurent Pessaint, Hanne Andersen, Mark G. Lewis, Alexander Wlodawer, James I. Mullins.

**Methodology:** Mahesh Agarwal, Xintao Hu, Santhi Devasundaram, Bhabadeb Chowdhury, Jenifer Bear, Robert Burns, Laurent Pessaint, Alexander Wlodawer, James I. Mullins, David J. Venzon.

**Project administration:** Barbara K. Felber.

**Resources:** Evangelos Terpos, Meletios Athanasios Dimopoulos.

**Software:** Duncan Donohue, David J. Venzon.

**Supervision:** George N. Pavlakis, Barbara K. Felber.

**Validation:** Margherita Rosati, George N. Pavlakis, Barbara K. Felber.

**Visualization:** Margherita Rosati, Barbara K. Felber.

**Writing – original draft:** Margherita Rosati, George N. Pavlakis, Barbara K. Felber.

**Writing – review & editing:** Margherita Rosati, Mahesh Agarwal, Xintao Hu, Santhi Devasundaram, Dimitris Stellas, Bhabadeb Chowdhury, Jenifer Bear, Robert Burns, Duncan Donohue, Laurent Pessaint, Hanne Andersen, Mark G. Lewis, Evangelos Terpos, Meletios Athanasios Dimopoulos, Alexander Wlodawer, James I. Mullins, David J. Venzon, George N. Pavlakis, Barbara K. Felber.

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
