## [Decision Letter · Decision Letter 0]

14 Aug 2021

Dear Dr. Felber,

Thank you very much for submitting your manuscript "Control of SARS-CoV-2 infection after Spike DNA or Spike DNA+Protein

co-immunization in rhesus macaques" for consideration at PLOS Pathogens. As with all papers reviewed by the journal, your manuscript was reviewed by members of the editorial board and by several independent reviewers. The reviewers appreciated the attention to an important topic. Based on the reviews, we are likely to accept this manuscript for publication, providing that you modify the manuscript according to the review recommendations.

Sincerely,

Shin-Ru Shih

Section Editor

PLOS Pathogens

Shin-Ru Shih

Section Editor

PLOS Pathogens

Kasturi Haldar

Editor-in-Chief

PLOS Pathogens

orcid.org/0000-0001-5065-158X

Michael Malim

Editor-in-Chief

PLOS Pathogens

orcid.org/0000-0002-7699-2064

Reviewer Comments (if any, and for reference):

Reviewer's Responses to Questions

**Part I - Summary**

Reviewer #1: Rosati et al. report a preclinical trials testing vaccines aimed at protecting from SARS-CoV-2 infection. They investigated two platforms, DNA alone and DNA+protein, that focused on the Spike protein included in different configurations as CoV-2 targeted antigen. The results indicated that while all vaccines provided protection for lower airways infection, the Spike DNA+ protein platform provided the best level of binding and neutralizing antibody and the strongest control of the infection. The vaccination follow up is carefully designed and the immunization evaluation is reported in detail. This study would benefit by is the evaluation of the immunity induced in macaques against the delta variant that is at present the most significant SARS-CoV-2 circulating variant.

Reviewer #2: Rosati M et al, report immunogenicity and protective efficacy of a DNA based SARS-CoV-2 targeted vaccine regimen expressing different pre-fusion stabilized Spike immunogens used via the I.M route of vaccination followed by electroporation in the rhesus macaque model. The authors build on their vast experience with pre-clinical studies with the DNA vaccine platform tested previously with HIV vaccine studies in the macaque model and specifically highlight the merits of combining DNA vaccines with protein+adjuvant (GLA-SE) co-delivered at the same anatomical site.

Overall, the article is very well written, the data is quite straightforward in the context of SARS-CoV-2 vaccine pre-clinical studies now extensively studied and documented. As a reviewer, I would argue that a technique with electroporation for mass vaccinations will potentially have less appeal in an ongoing pandemic only further complicated with a co-delivery of a protein vaccination with an adjuvant. Furthermore, the fact that mRNA vaccines have demonstrated the best-in-class immune responses (acute, while durability is still being investigated) and protective efficacies, the clinical translation of DNA vaccines in general and a co-delivery of DNA+Protein based immunization for SARS-CoV-2 looks challenging or less appealing at this stage of the pandemic. Having said that, like “all hands-on deck” when it comes to helping out with the pandemic, I submit that all vaccine platform approaches and data generated in pre-clinical studies, specifically with non-human primates (NHPs) are extremely useful in iterative design of future studies with other vaccine platforms. Based on this argument and the fact that the authors have done a really nice job in writing this article, I recommend publication of this work pending clarification of a couple of comments below.

**Part II – Major Issues: Key Experiments Required for Acceptance**

Reviewer #1: Considering that at present the most significant circulating variant is the delta variant and that the authors have tested the vaccine induced immunity against multiple variants that are less significant, this study would benefit by reporting in figure 3B the evaluation of the vaccine-induced immunity against the variant B.1.617.2. Obviously, it is easy to make a DNA vaccine that is specific for this variant, but it remains interesting to know how the immunity induced by a vaccine including a Spike sequence based on those used in human vaccination behaves against B.1.617.2 SARS-CoV-2.

Reviewer #2: 1. What was the criteria used to select treatment groups to be included in the challenge analyses?

a. One biggest disappointment for me a reviewer is the exclusion of group 5 from challenge. Group 5 had the best outcome with CD8+ T cell responses in comparison with group 2 which had higher Ab responses. The idea that induction of CD8+ T cell responses could benefit in the context of waning Ab based immunity would have been very interesting to document in the context of protective efficacy. Please discuss potential benefits of such a finding in potential future studies with heterologous challenge with SARS-CoV-2 variants.

2. The authors have done a good job in ensuring that Ab responses one week prior to challenge was correlated with protection outcome rather than peak values that highlights some sense of durability of immune responses. However, since most of the reporting is done in human trials for the peak 2 week time point after a vaccination regimen, I feel in Figure 4, it would be good to have a panel of plots highlighting potential correlates with 2 week Ab responses after the last vaccination for a side by side comparison with the durable Ab response correlate. This would highlight if there is differential waning of either binding, nAb or effector functions overall.

3. Do T cell responses either 2 weeks or 4 weeks after the challenge correlate with VLs in nasal swabs? Won’t it be good to also include that data since robust CD4+ T cell responses were observed ?

4. The authors discuss the potential advantages of a balanced humoral vs. cellular immunity afforded by DNA vaccines and here with DNA+Protein vaccines (lines 367-380). However, the critical observation is that the third vaccination could have really amplified these responses and in all fairness the emergency use approved mRNA vaccines use only 2 rounds of vaccinations. It remains to be seen whether the booster or the 3rd vaccination if approved will strongly amplify CD4+ and CD8+ T cell responses. So, at present, I would cautiously claim advantages of the DNA vaccine approach relative to mRNA vaccines in their ability to induce stronger cellular immunity.

5. Most journals and recent articles use the WHO nomenclature for identifying variants and would recommend that the authors refer to the variants referred to in the text and in figures as Alpha, Beta, Gamma variants….

**Part III – Minor Issues: Editorial and Data Presentation Modifications**

Reviewer #1: As the foldon is not present in the challenge virus, it may provide a tool to measure persistence of the immune response by evaluating antibodies against its sequence even after vial challenge. This might be an interesting piece of information for the vaccine platforms used in this study if the animals are still available and sufficient time has passed since the evaluation of protection against challenge.

Reviewer #2: Minor comments

1. Please indicate the # of animals in each treatment group in the macaque study on line 131 and also in Figure 1 as it is very difficult to get a good idea on the animal number.

2. Is titer computed as an AUC titer or endpoint? The authors are not consistent with terminology as the Y axis in Figure 1B says AUC and the results section on line 157 refer to endpoint titers followed by AUC titers on line 164. Please be consistent when detailing data.

PLOS authors have the option to publish the peer review history of their article (what does this mean?). If published, this will include your full peer review and any attached files.

Reviewer #1: No

Reviewer #2: No

Figure Files:

Data Requirements:

Reproducibility:

References:

---

## [Editor Report · Decision Letter 1]

7 Sep 2021

Dear Dr. Felber,

We are pleased to inform you that your manuscript 'Control of SARS-CoV-2 infection after Spike DNA or Spike DNA+Protein

co-immunization in rhesus macaques' has been provisionally accepted for publication in PLOS Pathogens.

Best regards,

Shin-Ru Shih

Section Editor

PLOS Pathogens

Shin-Ru Shih

Section Editor

PLOS Pathogens

Kasturi Haldar

Editor-in-Chief

PLOS Pathogens

orcid.org/0000-0001-5065-158X

Michael Malim

Editor-in-Chief

PLOS Pathogens

orcid.org/0000-0002-7699-2064
---

## [Editor Report · Acceptance letter]

17 Sep 2021

Dear Dr. Felber,

We are delighted to inform you that your manuscript, "Control of SARS-CoV-2 infection after Spike DNA or Spike DNA+Protein
co-immunization in rhesus macaques," has been formally accepted for publication in PLOS Pathogens.

Best regards,

Kasturi Haldar

Editor-in-Chief

PLOS Pathogens

orcid.org/0000-0001-5065-158X

Michael Malim

Editor-in-Chief

PLOS Pathogens

orcid.org/0000-0002-7699-2064